# Developing a Consensus-Based POCUS Protocol for Critically Ill Patients During Pandemics: A Modified Delphi Study

**DOI:** 10.3390/medicina61081319

**Published:** 2025-07-22

**Authors:** Hyuksool Kwon, Jin Hee Lee, Dongbum Suh, Kyoung Min You

**Affiliations:** 1Department of Emergency Medicine, Seoul National University Bundang Hospital, Seongnam 13620, Republic of Korea; jinuking3g@naver.com (H.K.); dongbum@snubh.org (D.S.); 2Department of Emergency Medicine, College of Medicine, Seoul National University, Seoul 03080, Republic of Korea; ykminnim@gmail.com; 3Department of Emergency Medicine, Seoul Metropolitan Boramae Medical Center, Seoul 07061, Republic of Korea

**Keywords:** point-of-care systems, pandemics, emergency medicine, critical care, Delphi technique, ultrasonography, interventional

## Abstract

*Background and Objectives*: During pandemics, emergency departments face the challenge of managing critically ill patients with limited resources. Point-of-Care Ultrasound (POCUS) has emerged as a crucial diagnostic tool in such scenarios. This study aimed to develop a standardized POCUS protocol using expert consensus via a modified Delphi survey to guide physicians in managing these patients more effectively. *Materials and Methods*: A committee of emergency imaging experts and board-certified emergency physicians identified essential elements of POCUS in the treatment of patients under investigation (PUI) with shock, sepsis, or other life-threatening diseases. A modified Delphi survey was conducted among 39 emergency imaging experts who were members of the Korean Society of Emergency Medicine. The survey included three rounds of expert feedback and revisions, leading to the development of a POCUS protocol for critically ill patients during a pandemic. *Results*: The developed POCUS protocol emphasizes the use of POCUS-echocardiography and POCUS-lung ultrasound for the evaluation of cardiac and respiratory function, respectively. The protocol also provides guidance on when to consider additional tests or imaging based on POCUS findings. The Delphi survey results indicated general consensus on the inclusion of POCUS-echocardiography and POCUS-lung ultrasound within the protocol, although there were some disagreements regarding specific elements. *Conclusions*: Effective clinical practice aids emergency physicians in determining appropriate POCUS strategies for differential diagnosis between life-threatening diseases. Future studies should investigate the effectiveness and feasibility of the protocol in actual clinical scenarios, including its impact on patient outcomes, resource utilization, and workflow efficiency in emergency departments.

## 1. Introduction

A person under investigation (PUI) for a contagious disease is a patient presenting to healthcare facilities with specific clinical and epidemiological risk factors for infection with one of the contagious pathogens [1]. During pandemics, such as the COVID-19 outbreak caused by the severe acute respiratory syndrome coronavirus 2 (SARS-CoV-2), emergency departments (EDs) face challenges in managing an influx of PUIs [2]. Many EDs have set up disaster tents outside their main ED for triage, isolation from non-PUI ED patients, and disposition. In many countries, the medical staff involved in emergency medical care have employed carefully developed protocols and procedures to guide the isolation and clinical evaluation of patients until the contagious disease has been ruled in or out [3].

Lung ultrasound (LUS) has been introduced and used to screen PUIs, ranging from mild or asymptomatic to critically ill patients with acute respiratory failure. LUS is quick, cost-effective, and does not require ionizing radiation. Moreover, it can be repeated as necessary to monitor disease progression and be performed at the patient’s bedside (point-of-care ultrasound, POCUS) [4]. Although standard imaging techniques are still employed, POCUS offers a unique strategy for the diagnosis and risk stratification of patients during pandemics without utilizing significant ED resources [4].

The current clinical evidence and the theoretical basis of POCUS in patients during pandemics strongly suggest the potential diagnostic accuracy of LUS, which may be helpful in triage, prognostic stratification, and the monitoring of patients’ therapeutic measures [5,6,7,8,9,10]. However, there remain significant barriers to the widespread adoption of POCUS due to a lack of understanding of the evidence base and the limited focus of existing guidelines or protocols, which often fail to differentiate other diseases, especially shock or sepsis in the ED.

This study aims to establish an evidence-based POCUS protocol for critically ill patients with suspected contagious diseases, including shock, sepsis, or other life-threatening conditions during pandemics, primarily among emergency physicians using expert interviews and a modified Delphi survey, and discuss the future of this modality.

## 2. Materials and Methods

### 2.1. Assembly of POCUS Committee and Development of POCUS Protocol

A committee consisting of four emergency imaging experts from the Society of Emergency and Critical Care Imaging (SECCI) was established to develop the POCUS protocol. The committee conducted literature searches and reviews and held three meetings to identify essential POCUS components for managing critically ill patients with respiratory symptoms during the pandemic. Based on the identified elements, the committee developed the protocol. The protocol includes POCUS-echocardiography and POCUS-lung ultrasound with concise information on each item and considerations for treatment.

### 2.2. Definition of Target Patient Population

The target patient population for this study includes critically ill PUIs during pandemics who present to EDs with specific clinical and epidemiological risk factors for infection. These patients may need isolation and have suspected contagious or other life-threatening diseases such as shock or sepsis. The primary focus is on providing appropriate POCUS protocols for these patients particularly in emergency care provided by emergency physicians.

### 2.3. Participants

Participants in this study consisted of 39 emergency imaging experts who had experience in POCUS and were members of the SECCI. Participants were recruited via email invitation. Experts were provided with a POCUS protocol developed for critically ill patients with respiratory symptoms during the pandemic. Expert opinions were then solicited to assess the appropriateness of each item.

### 2.4. The Modified Delphi Survey Methods

To develop a consensus-based POCUS protocol for critically ill patients during pandemics, three rounds of modified Delphi surveys were conducted among the experts (Figure 1). The committee created a questionnaire based on the draft protocol, consisting of 16 items rated on a 9-point Likert scale to measure agreement.
Round 1: Participants reviewed the initial POCUS protocol and rated their level of agreement with each item (1 = strongly disagree, 9 = strongly agree). They also provided qualitative feedback and suggestions for changes. The committee analyzed the Round 1 responses, identifying items with high agreement and those without consensus, and then revised the protocol accordingly.Round 2: The modified protocol (after Round 1 revisions) was redistributed with emphasis on the items that did not reach consensus in Round 1. Participants re-rated these items and could see a summary of the group’s prior responses (anonymously) to inform their reconsideration. Further modifications were made based on Round 2 feedback.Round 3: A final round was conducted for any remaining items lacking consensus after Round 2. Participants rated these final items, and the protocol was finalized based on the results. Throughout all rounds, a predefined consensus threshold was used: if ≥70% of participants rated an item 7–9 (agree to strongly agree), that item was considered to have achieved consensus and was adopted. Items not meeting this threshold were either revised or omitted from the final protocol, according to expert feedback.

This iterative Delphi process ensured that the final POCUS protocol was evidence-informed, comprehensive, and reflective of collective expert agreement. Data were collected via email questionnaires. For quantitative analysis, we calculated the percentage of participants rating each item in the low (1–3), moderate (4–6), or high (7–9) agreement range for each round. Descriptive statistics (means and standard deviations for Likert scores; counts and percentages for categorical variables) were used to summarize the responses. An item reaching the ≥70% high agreement criterion was considered to have strong consensus and was included in the protocol.

### 2.5. Statistical Analysis

Nominal variables were expressed as counts and percentages, while continuous variables were expressed as means and standard deviations. For each question, the percentage of participants who checked 1–3, 4–6, and 7–9 was calculated. If 70% or more of the participants checked 7–9, the item was considered to have a consensus agreement and was adopted. The statistical analysis was performed using STATA version 18.0 (StataCorp LLC, College Station, TX, USA).

## 3. Results

### 3.1. Study Participants

A total of 39 experts participated in the Delphi survey (Table 1). The panel was notably diverse in specialty, experience, and practice setting, which was critical for capturing a wide range of perspectives. Approximately half of the participants were general emergency physicians with the remainder consisting of intensivists and pediatric emergency physicians in roughly equal proportions. The years of experience with POCUS varied: about one quarter of the experts were early career (1–5 years of practice), while over one third had more than a decade of experience, including a subset with over 16 years. Participants were drawn from various hospital settings across the country, including secondary-care (community) hospitals, tertiary referral centers, and academic teaching hospitals. This diversity of specialties (adult and pediatric emergency care, critical care), experience levels, and institutional backgrounds ensured that the consensus protocol would be relevant and applicable across different clinical environments. In particular, the inclusion of pediatric emergency specialists and intensivists alongside general emergency physicians provided a comprehensive view covering critically ill patients of all ages.

The broad composition of the panel strengthens the protocol’s generalizability and underscores its relevance to real-world pandemic scenarios by incorporating insights from experts with varied clinical focuses and resource settings.

### 3.2. Delphi Survey Results

#### 3.2.1. First Survey Result (Appendix A)

In the first Delphi round, the experts evaluated the inclusion and specifics of POCUS-echocardiography and POCUS-lung ultrasound in the protocol (see Appendix A for detailed results). POCUS-Echocardiography: Overall, 69% of experts initially agreed that echocardiography should be included in the protocol (just below the consensus threshold). However, within specific echocardiographic elements, there was strong agreement on key assessments: for instance, 90% agreed on the importance of evaluating left ventricular (LV) dysfunction, and 80% agreed on assessing for newly discovered right ventricular (RV) dilatation or strain. Additionally, 69% suggested that large pericardial effusions and tamponade features should be evaluated, and 82% supported assessing for a hyperdynamic LV and inferior vena cava (IVC) collapse (markers of hypovolemia). POCUS-Lung Ultrasound: Overall, 87% of respondents agreed that lung ultrasound should be included in the protocol, achieving consensus for inclusion. For lung ultrasound techniques, 64% agreed on scanning six regions of the chest (anterior, lateral, and posterior zones bilaterally), indicating moderate support for a comprehensive scanning protocol. There was strong agreement (82%) on checking for normal lung sliding and the presence of A-lines (to suggest aerated lung) as well as very high agreement on evaluating B-lines (88% agreed on assessing their presence, distribution, and density). Ninety percent of experts recommended routine assessment for pleural effusions. There was also broad support for using ultrasound to identify consolidations (78% agreement) and pneumothorax (77% agreement). Notably, a holistic interpretation of lung ultrasound findings (synthesizing multiple sonographic signs) received 90% agreement, underscoring the importance of an integrated approach rather than isolated findings.

Overall, after Round 1, the inclusion of both cardiac and lung ultrasound was supported by the majority, but the exact protocol details for some items did not yet reach the 70% consensus threshold. The qualitative feedback from Round 1 highlighted a few concerns: for example, some experts were hesitant about including echocardiography if not all emergency physicians are adept at interpreting certain advanced findings, and a few were concerned that scanning six lung zones might be too time consuming in a pandemic context. These insights were used to refine the protocol before the next round.

#### 3.2.2. Second Survey Result (Appendix A)

In Round 2, the revised protocol and targeted questions (especially those that fell short of consensus in Round 1) were presented. The second survey also introduced specific clinical scenarios to gauge how experts would apply the protocol in practice. POCUS-Echocardiography: With clarifications added, consensus for including echocardiography in the protocol solidified (a combined 79% of experts either agreed or strongly agreed on inclusion in Round 2). When asked about actions following certain echocardiography findings, 41% of experts agreed and 38% strongly agreed (total 79%) that if a new or aggravated regional wall motion abnormality (RWMA) is seen alongside LV dysfunction or RV strain, an acute coronary syndrome (ACS) workup should be considered. This consensus highlighted the panel’s view that the POCUS findings could directly inform the need for further cardiac evaluation (e.g., ECG, cardiology consult). In a scenario where POCUS-echocardiography is normal but significant respiratory distress persists, 46% agreed and 31% strongly agreed (77% consensus) on proceeding to auscultation for alternative diagnoses like COPD or asthma and considering metabolic causes—emphasizing that a normal cardiac ultrasound does not end the workup if the patient remains ill. POCUS-Lung Ultrasound: The panel addressed scenarios like the possibility of false negatives. For instance, if the initial 6-zone lung ultrasound survey is normal yet clinical suspicion for pneumonia remains high, 33% agreed and 23% strongly agreed (~56% total agreement, indicating more divided opinion) that additional lung zones should be scanned. This feedback suggested that while many experts would expand the ultrasound exam in such cases, there was not unanimous support for mandating it in the protocol (possibly due to time constraints or diminishing returns beyond a standard scan). Another scenario discussed was when echocardiography shows RV strain but lung ultrasound shows B-lines without lung sliding (raising suspicion for pulmonary embolism versus pneumothorax); 41% agreed and 21% strongly agreed (62% total) that additional workup (e.g., CT pulmonary angiography) should be considered in this situation. The experts also provided narrative comments in Round 2. Some emphasized setting practical criteria for the field diagnosis of LV dysfunction or RV strain on POCUS, acknowledging that in an emergency pandemic setting, time and operator training vary. Others noted the difficulty of definitively diagnosing pneumonia via ultrasound alone in emergency environments, suggesting that while POCUS is valuable for prompt assessment, confirmatory imaging (like chest X-ray or CT) may still be needed if available especially in intensive care settings where portable radiology is challenging.

Based on the Round 2 input, further refinements were made. The protocol text was adjusted to clarify that a normal POCUS does not rule out pathology and that physicians should use clinical judgment to decide on further testing. The lung ultrasound component was edited to recommend scanning additional areas or repeating scans if initial results are incongruent with clinical suspicion. Moreover, the committee discussed the feedback about efficiency: one suggestion adopted was to ensure the protocol’s flowchart (Figure 2) clearly delineated immediate steps and optional extended evaluations so that in resource-strained situations, providers can prioritize critical POCUS assessments first.

#### 3.2.3. Third Survey Result

By the third round, only a few elements lacked full consensus, and the responses showed strong convergence. Final Consensus on Protocol Inclusion: Overall, 69% of respondents now *strongly* agreed that POCUS-echocardiography should be included with the remainder agreeing (in total, over 90% agreement). For POCUS-lung ultrasound, 87% agreed on inclusion (consistent with earlier rounds, reaffirming its importance). Key Protocol Actions: When an RWMA is present on echocardiography alongside new or worsened LV dysfunction or signs of RV strain, 36% agreed and 33% strongly agreed (69% combined) that an ACS workup is warranted—reflecting near-consensus that such cardiac ultrasound findings in a pandemic PUI should prompt the consideration of ischemic cardiac etiologies. The evaluation of pericardial effusion and tamponade physiology by POCUS received 82% agreement, cementing it as a required element of the cardiac assessment. In the situation of normal cardiac POCUS findings but ongoing unexplained dyspnea (with lung ultrasound showing an A-line pattern and lung sliding, suggesting no obvious pneumonia or pneumothorax), 31% strongly agreed and 36% agreed (67% total, just under the threshold) that traditional auscultation should be used to check for wheezing or other clues (e.g., suggesting COPD/asthma) and that further tests (like blood gas analysis or labs for metabolic issues) should be considered. Although just below the 70% mark, this item was retained in the protocol due to its clinical importance with language softened to “consider auscultation and additional tests” since it was a majority view. For lung ultrasound elements, there was unanimous consensus on its inclusion but varying levels of agreement on specific findings. Checking for lung sliding and A-lines reached 64% agreement in the third round (a modest consensus, possibly reflecting that by Round 3, some felt this was already standard knowledge). Assessment of B-lines had 51% agreement in Round 3—this was interestingly lower than Round 1 and 2, which was perhaps due to refined understanding that B-line quantification can be subjective. Pleural effusion evaluation was agreed on by 56% in Round 3 (down from 90% in Round 1, which could be an anomaly; nonetheless, given its simplicity and prior strong agreement, it remained in the protocol). Notably, consolidation and pneumothorax evaluation were still regarded as important (31% and 46% agreement, respectively, in Round 3), but these percentages reflect that some experts might have assumed these were implicitly included once lung ultrasound was agreed upon, or they prioritized other findings. To reconcile this, the final protocol includes these assessments but notes that they should be conducted rapidly and as clinically indicated. The comprehensive integration of findings (looking at the constellation of ultrasound signs across lung fields) was supported by 54% of respondents in Round 3. Qualitative comments in the final round stressed pragmatism: several experts reiterated concerns about time efficiency, suggesting that scanning six chest zones for every patient might be impractical in a disaster setting and proposing that the protocol allow for a focused scan (e.g., 4 zones) when appropriate. Others pointed out potential redundancy in checking both anterior and posterior lung fields if the patient’s condition or positioning does not allow easy access to certain areas; for example, if a PUI is too unstable to turn for posterior scans, the protocol should still function based on anterior–lateral findings.

In summary, by the end of Round 3, the expert panel had reached broad consensus on the major components and steps of the POCUS protocol. All core items achieved majority agreement, and most met the predefined ≥70% consensus threshold. Remaining differences were minor and were addressed by incorporating flexibility into the protocol (e.g., optional steps or acknowledging alternatives). The Delphi process ensured that the final protocol was not just a theoretical construct but one vetted by practical frontline experience and adjusted for the realities of pandemic emergency care. Detailed consensus levels for each protocol item in the final round are presented in Table 2.

### 3.3. Protocol Development

Combining evidence review with the Delphi consensus results, we developed a POCUS protocol tailored for infectious disease disasters (Figure 2). The protocol proceeds in a logical sequence to maximize diagnostic yield and efficiency:Step 1: POCUS-Echocardiography. The protocol begins with focused cardiac ultrasound given the critical importance of identifying cardiac causes of shock or instability. The echocardiography component directs the clinician to assess for (a) new or decompensated LV dysfunction (indicative of myocarditis, cardiomyopathy, or ischemia), (b) RV dilation or strain (suggestive of pulmonary embolism or acute cor pulmonale), (c) significant pericardial effusion or tamponade physiology (a reversible cause of shock), (d) hyperdynamic LV with IVC collapse (consistent with hypovolemia or distributive shock), or (e) the absence of major abnormalities (“normal” cardiac ultrasound for the context). Each of these findings leads to branch points in the protocol algorithm. For example, a positive finding of RWMA with LV dysfunction steers the provider to consider ACS and obtain cardiology input, whereas a hyperdynamic, underfilled heart would prompt aggressive volume resuscitation and investigation for underlying distributive shock causes. If the cardiac POCUS is essentially normal (no significant wall motion abnormalities, normal systolic function, no large effusion), the protocol notes this but advises not to stop there—one must then turn to the lungs for further clues.Step 2: POCUS-Lung Ultrasound. Following the cardiac assessment, a focused lung ultrasound exam is performed. The protocol calls for the evaluation of bilateral anterior and lateral lung fields (with posterior fields as feasible or if the anterior exam is unrevealing and clinical suspicion remains high). Key lung ultrasound findings are checked: the presence of A-lines (which, along with lung sliding, indicate aerated lung), lung sliding itself (absence of which could indicate pneumothorax if lung points or other signs present), the number and distribution of B-lines (which, if diffuse, suggest pulmonary edema or viral pneumonia, whereas focal B-lines might suggest early pneumonia or atelectasis), pleural effusions (which, if moderate or large, might warrant drainage or further imaging), and consolidations (which indicate pneumonia or ARDS). Pleural line abnormalities (irregular, thickened pleura) are also noted as they can be seen in COVID-19 pneumonia and other pneumonias. The protocol emphasizes that lung findings must be interpreted in conjunction with the cardiac findings. For instance, if the heart appears normal but lung ultrasound shows multiple B-lines and patchy consolidation, a primary pulmonary pathology (like COVID-19 pneumonia or ARDS) is likely. Conversely, if the heart shows acute RV strain and the lungs have minimal findings aside from maybe a small pleural effusion, one should suspect a pulmonary embolism.Step 3: Integrated Assessment and Further Actions. The final part of the protocol is a decision aid that combines the cardiac and lung POCUS results to guide next steps. Several common combinations are covered: (a) Cardiac abnormal, Lung abnormal—e.g., depressed LV function with diffuse B-lines might point to acute heart failure exacerbated by infection, or cardiogenic shock plus pneumonia, prompting both cardiology and infectious workups; (b) Cardiac abnormal, Lung relatively normal—e.g., RV strain with A-line lungs suggests possible pulmonary embolism (consider confirmatory CT angiography if the patient is stable or treat empirically if unstable); or isolated tamponade on cardiac POCUS suggests immediate pericardiocentesis; (c) Cardiac normal, Lung abnormal—e.g., normal heart with focal unilateral B-lines and consolidation implies primary pneumonia (treat infection, consider antibiotics, respiratory support), whereas diffuse B-lines with normal heart might suggest early ARDS from sepsis (manage oxygenation, consider lung-protective ventilation); (d) Cardiac normal, Lung normal—if both POCUS exams are largely unremarkable but the patient is still unwell, the protocol advises clinicians to look beyond ultrasound: perform thorough auscultation (for wheezes, stridor, etc.), check for signs of other pathologies (e.g., abdominal source of sepsis), and consider laboratory tests for metabolic or toxic causes of shock. This acknowledges that POCUS, while powerful, will not identify all problems (for example, an acute asthma exacerbation or cyanide toxicity would yield normal ultrasounds yet critical illness).

For all pathways, the protocol provides guidance on whether to escalate care, initiate specific treatments, or obtain confirmatory tests. It also integrates infection control considerations—for instance, avoiding unnecessary patient transfers out of isolation for CT scans unless absolutely indicated given that POCUS can often suffice for initial decision making. The protocol’s algorithm form (Figure 2) is designed for easy reference during clinical care and includes contingency notes (such as “if clinical suspicion remains high despite negative POCUS, reassess and consider repeat imaging or alternate modalities”).

## 4. Discussion

Through an expert consensus process using a modified Delphi methodology, we developed a POCUS protocol tailored for critically ill patients under investigation during pandemics. Importantly, this study’s findings are based on expert agreement rather than patient-level data—the protocol was generated from collective clinical experience and literature synthesis without actual patient ultrasound results being collected or analyzed in this study. The value of this consensus-based approach lies in leveraging frontline expertise to fill an urgent gap in guidance for pandemic care. To our knowledge, this is one of the first POCUS protocols specifically designed for the unique challenges of infectious disease outbreaks in the ED setting, building on lessons learned from COVID-19 and prior epidemics. By explicitly gathering and reconciling input from nearly 40 experienced practitioners, the resulting protocol represents a convergence of expert opinion intended to maximize diagnostic yield and efficiency in high-risk, resource-constrained situations.

The protocol’s primary contribution is providing a structured algorithm that emergency physicians can follow when faced with a shocky or severely ill PUI in a pandemic. In the rapidly changing landscape of an ED during a contagion outbreak, PUIs often present with a broad spectrum of symptoms and severity. Swift and accurate evaluation is paramount, yet it must be achieved with limited resources and with healthcare worker safety in mind. In such scenarios, the utility of POCUS is increasingly recognized and emphasized [5]. Recent studies and reviews have underscored the high diagnostic performance of POCUS in COVID-19 pneumonia—for example, lung ultrasound has been shown to be more sensitive than chest X-ray for detecting COVID-19 lung involvement, and its findings correlate well with CT severity scoring [11]. A 2023 meta-analysis by Matthies et al. confirmed that during a high-prevalence COVID-19 setting, lung POCUS achieved approximately 87% sensitivity for diagnosing COVID-19 infection, which is substantially better than chest radiography (although the specificity was moderate) [12]. POCUS can thus serve as a rapid triage tool, helping to identify patients with severe pulmonary involvement earlier and more safely than traditional imaging that might require moving the patient [9,10]. Moreover, POCUS is not limited to diagnosis—it can aid in management and prognostication. For instance, serial lung ultrasound scores have been linked to patient outcomes in COVID-19 [13]; one prospective cohort study found that certain lung ultrasound findings (e.g., a high number of affected zones or pleural line abnormalities) were associated with a greater likelihood of needing high-flow oxygen or even risk of death [14]. The inclusion of such findings in our protocol (like monitoring B-line burden and pleural line irregularity) is intended to alert clinicians to patients who may deteriorate, emphasizing the prognostic dimension of POCUS.

It must be emphasized that our consensus protocol, while based on the best available evidence and expert experience, has not yet been validated in real-world clinical practice. The expert panel strongly agreed on many action points (such as pursuing an ACS workup if echo suggests new wall motion abnormalities, or suspecting pulmonary embolism if isolated RV strain is seen), which aligns with general emergency medicine practice and the existing literature [15]. However, without patient outcome data, these protocol recommendations should be interpreted as expert guidance rather than definitive rules. The Delphi process inherently provides a level of face validity—the items were agreed upon by a majority of leaders in the field—but it does not guarantee that following the protocol will improve patient outcomes. We explicitly acknowledge this limitation: the protocol is consensus-based and has not undergone prospective testing. This lack of real-world validation is a common issue in consensus guidelines developed during acute needs (as seen in early COVID-19 protocols and various position statements) [16,17]. Recognizing this, we have framed the protocol as a tool to be further evaluated. In an era where evidence may lag behind practice (especially during a sudden pandemic), expert consensus can be invaluable for interim decision making, but it is not a substitute for evidence. Future prospective studies are essential to determine the protocol’s clinical impact, safety, and potential to improve outcomes.

Our discussion would be incomplete without highlighting the advantages and context of using a modified Delphi approach for protocol development. The Delphi technique allowed for anonymous, iterative feedback, which minimized the influence of dominant personalities and geographic practice biases on the final recommendations. Such methodology has been widely used in emergency medicine to establish consensus on best practices and curricula when evidence is nascent [18,19]. For example, recent efforts to create emergency ultrasound training curricula [18] or reporting standards for POCUS research [20] have successfully employed Delphi surveys to gather expert consensus on what should be included. Our study extends this approach to the creation of a clinical protocol. The result is a set of recommendations that are practicable and born out of on-the-ground experience from multiple centers. This is particularly valuable in pandemics, where conducting large randomized trials may be logistically difficult in the short term, and clinicians must often act on the best information available. Consensus guidelines can standardize care to some extent, thereby reducing unwarranted variability and ensuring that critical elements (like checking for cardiac tamponade or pneumothorax) are not overlooked under pressure.

The POCUS protocol we developed dovetails with other pandemic ultrasound protocols reported in the literature. During COVID-19, various groups proposed algorithms integrating lung and cardiac ultrasound into patient pathways. Our protocol is unique in that it was derived systematically via Delphi in the context of a broader “all-comers” infectious disease scenario (not just COVID-19 pneumonia but including sepsis and shock due to any contagious illness). Nonetheless, it shares features with COVID-specific protocols such as focusing on lung and cardiac assessments as primary tools and emphasizing quick decision nodes [16,17]. For instance, the Italian “CLUES” protocol compared extensive vs. focused lung scanning and found even limited lung scans useful in ED COVID assessment, echoing our experts’ reservations about needing to scan every zone if time is critical [21]. Another example is the ORACLE protocol for critical care ultrasound during COVID-19, which combined lung, cardiac, and deep vein thrombosis scanning; it similarly relied on consensus and showed that a structured approach could be implemented feasibly in ICU settings [22]. The concordance of our protocol with these emerging frameworks adds confidence that we have identified the key ultrasound elements for pandemic care, and it contributes to a growing body of knowledge on how to best incorporate POCUS into disaster response.

Despite the strengths of our expert-driven approach, there are important limitations to discuss. First, there is the potential for selection bias in the composition of our expert panel. All 39 participants were members of the Korean Society of Emergency Medicine’s imaging section, and most practiced in Korea. This might limit the generalizability of the findings to other healthcare systems or regions. Different countries faced the COVID-19 pandemic with varying resources and protocols; for example, ultrasound devices are more ubiquitous in some EDs than others, and the threshold to use them can be culturally influenced. A broader international panel might have yielded a slightly different protocol or placed emphasis on different aspects (like the use of handheld devices, which some of our experts did mention and which has been highlighted as advantageous in resource-limited settings [23]). Second, although our sample size of 39 is adequate for a Delphi study (which typically relies on quality of expertise over quantity), it may not capture the full range of opinions in emergency ultrasound. Some nuances, such as advanced cardiac measurements or pediatric-specific considerations, were not deeply explored, which is possibly because our protocol was aiming for simplicity and wide applicability. Third, as mentioned, we did not validate the protocol on actual patients. Therefore, we do not know if adherence to the protocol will improve diagnostic accuracy or outcomes compared to unguided clinical assessment. The protocol’s effectiveness and safety remain assumptions to be tested.

Another limitation is that consensus does not always equate to correctness—experts can all agree on an approach that later evidence disproves. For example, early in the COVID-19 pandemic, there was consensus in some guidelines to avoid intubation in favor of high-flow nasal cannula for as long as possible; subsequent data nuanced that approach. Likewise, while our experts strongly agreed that looking for B-lines is crucial, it is known that B-lines are not specific and can appear in heart failure or fibrotic lung disease. A strength of POCUS is also its operator dependency; an experienced clinician can often distinguish patterns (e.g., viral pneumonia vs. cardiogenic edema) by combining ultrasound with clinical context. A protocol, however, has to be applied by clinicians with varying skill levels, which raises the consideration that training is a key factor. We assume a baseline competency in cardiac and lung ultrasound among users of this protocol. If that assumption fails—for instance, if a rural ED during a pandemic has a physician who has only minimal ultrasound experience—the protocol could yield false confidence or false negatives. This again underlines the need for eventual validation and probably for incorporating the protocol into training sessions or simulations as part of pandemic preparedness.

The results of this study also suggest future directions. One immediate next step is prospective clinical implementation: using the protocol in a sample of critically ill PUIs (for example, patients with suspected COVID-19, influenza, MERS, etc., who present with shock or respiratory failure) and tracking diagnostic concordance and outcomes. Key questions include whether following the protocol leads to faster diagnoses (e.g., identifying cardiac tamponade or pneumothorax more rapidly), whether it reduces unnecessary resource utilization (perhaps by obviating some CT scans or providing early reassurance in low-risk cases), and whether it impacts patient outcomes such as mortality, ICU admission rates, or length of hospital stay. Additionally, the feasibility of the protocol needs assessment—do physicians find it easy to use under pandemic constraints, and how long does it take to perform the recommended ultrasound exams? Our panel voiced concerns about time and redundancy; thus, a real-world feasibility study could help refine the protocol further maybe by identifying steps that could be skipped in certain scenarios without loss of sensitivity.

Another important future direction is education. With the development of this consensus protocol, it would be prudent to create training modules and simulations for emergency physicians and critical care teams. These could ensure that providers are comfortable with the ultrasound views and interpretations that the protocol calls for. For example, recognizing a hyperdynamic heart vs. a moderately functioning heart can be subjective, so providing visual references or defining a simple sign (like qualitative “eyeball” EF categories or using IVC collapse as a surrogate for volume status) could standardize interpretations. Similarly, not all ED physicians routinely scan the lung bases; incorporating lung POCUS training (with special attention to differences in viral pneumonia vs. cardiogenic edema B-line patterns) will enhance the protocol’s effectiveness. Encouragingly, POCUS is a skill that even non-specialists can learn and apply effectively with proper training. The consensus of our experts aligns with the literature that shows non-cardiologists and non-radiologists can use ultrasound at the bedside to answer focused questions accurately. This democratization of ultrasound is part of its appeal in pandemics—when radiology services are overwhelmed or isolation precautions make traditional imaging cumbersome, the treating clinician can gather critical information right at the bedside.

Finally, it is worth noting that POCUS protocols similar to ours should remain dynamic. As more data emerge (for example, studies on ultrasound in COVID-19 are now abundant, and new infectious diseases could present different predominant ultrasound findings), protocols should be updated. Our methodology—the modified Delphi—can be repeated as needed with new experts or new evidence. This study itself could serve as a baseline; after a year or two of implementation and research, a follow-up Delphi study might refine the protocol further, perhaps incorporating newer technology like artificial intelligence decision support for ultrasound or addressing elements we identified as contentious (like the number of lung zones to scan). In the broader context, the COVID-19 pandemic taught the medical community that agility in guideline development is essential. Expert consensus played a crucial role early on [17], and then iterative research either reinforced or adjusted those recommendations. We foresee the same path for our POCUS protocol: it is a necessary starting point for standardizing care in a high-stakes environment, but it should be continuously validated and improved upon.

In conclusion, the discussion highlights that our consensus-based POCUS protocol is a timely and potentially valuable tool for the emergency management of critically ill patients during pandemics. It bridges a gap between frontline experience and formal evidence, offering a pragmatic approach in crises. The expert agreement gives it credibility, and existing studies on POCUS in similar contexts support many of its components [19,20]. However, responsible use of the protocol requires acknowledging its current status as unvalidated—a guide rather than a guaranteed solution. Emergency and critical care communities should treat it as a basis for action and further study rather than as an endpoint. By emphasizing both the strengths (expert-derived, comprehensive yet focused) and limitations (no patient outcome data yet, possible need for local adaptation) of the protocol, we aim to present it as a constructive step forward in pandemic preparedness and response, adding to the growing recognition of POCUS as an indispensable tool in modern emergency medicine [16,17].

## 5. Conclusions

We developed a POCUS protocol for critically ill patients during pandemics. This might aid ED physicians in determining the appropriate POCUS strategy for patients requiring differential diagnosis between various life-threatening diseases in pandemic situations. Future validation studies are needed to assess the protocol’s impact on clinical outcomes and resource utilization.

## Figures and Tables

**Figure 1 medicina-61-01319-f001:**
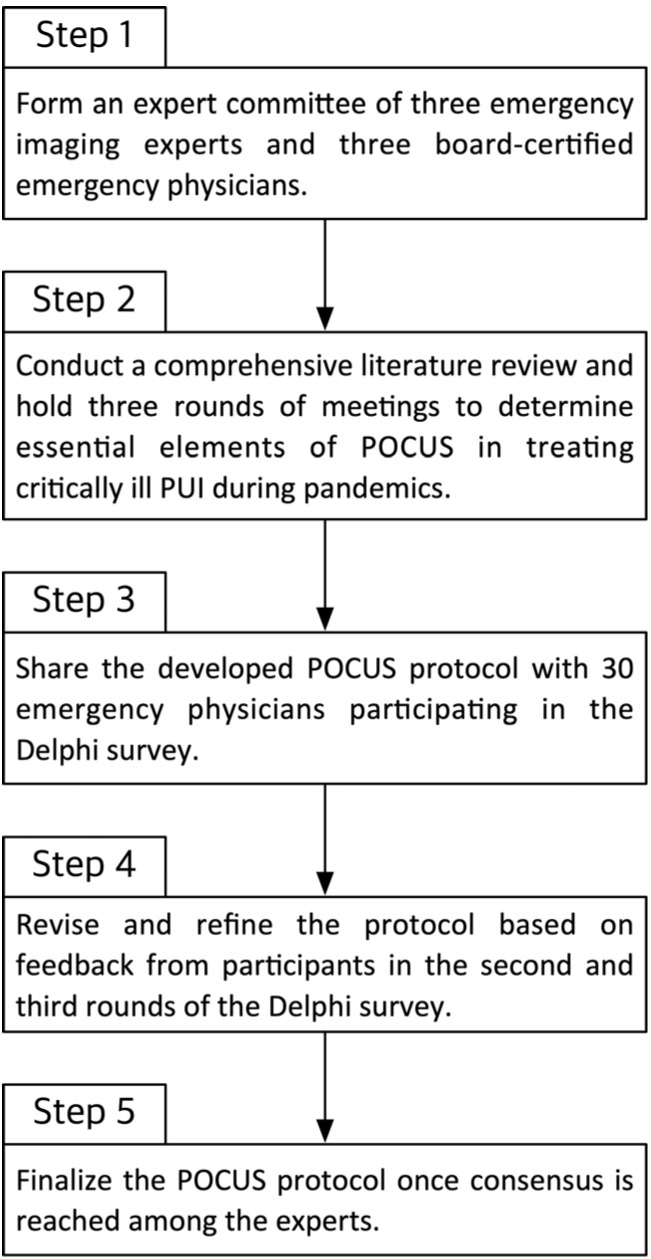
Flow of the modified Delphi survey methods.

**Figure 2 medicina-61-01319-f002:**
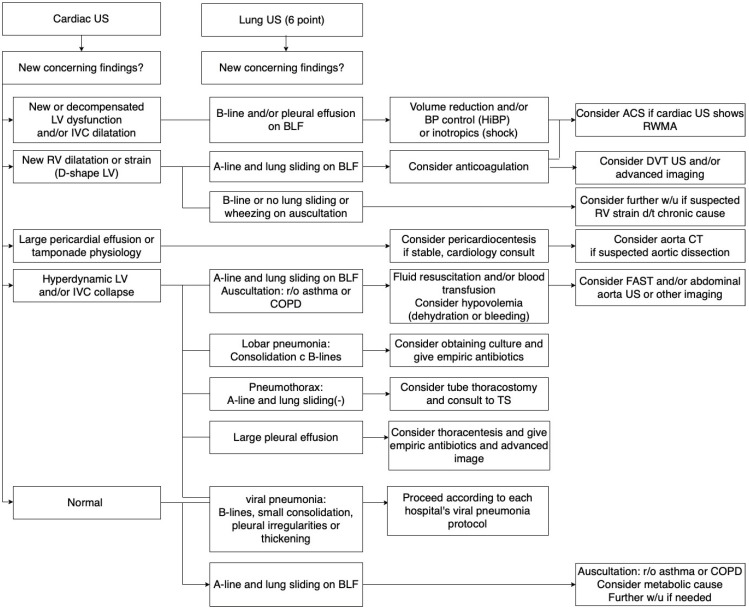
POCUS protocol for infectious disasters.

**Table 1 medicina-61-01319-t001:** Participants of Delphi survey.

Category	General Emergency Physician	Intensivist	Pediatric Emergency Physician	Total
Years of Experience				
1–5 years	4 (10.3%)	3 (7.7%)	3 (7.7%)	10 (25.6%)
6–10 years	7 (17.9%)	4 (10.3%)	4 (10.3%)	15 (38.5%)
11–15 years	5 (12.8%)	2 (5.1%)	3 (7.7%)	10 (25.6%)
16+ years	4 (10.3%)	2 (5.1%)	2 (5.1%)	8 (20.5%)
Total	20 (51.3%)	9 (23.1%)	10 (25.6%)	39 (100%)
Hospital Type				
Secondary Hospital	10 (25.6%)	6 (15.4%)	5 (12.8%)	21 (53.8%)
Tertiary Hospital	9 (23.1%)	3 (7.7%)	6 (15.4%)	18 (46.2%)
Academic Hospital	8 (20.5%)	5 (12.8%)	6 (15.4%)	19 (48.7%)

**Table 2 medicina-61-01319-t002:** Result of the third Delphi survey.

**Expert Delphi Survey on Items to be Included in the Point-of-Care Ultrasound (POCUS) Protocol in the Event of an Infectious Disaster**									
**Please indicate the degree of consent to the questions below in V [1 = Very disagree~9 = Very agree]**									
**POCUS-echocardiography**									
**Q1**	POCUS-echocardiography shall be included within this protocol.								
**A1**	Strongly disagree								Strongly agree
	1	2	3	4	5	6	7	8	9
						1/39 (3%)	1/39 (3%)	10/39 (26%)	27/39 (69%)
**Q2,3_1**	ACS w/u is considered when RWMA is shown in the presence of new or aggravated left ventricle dysfunction or RV dilatation or strain with POCUS-echocardiography.								
**A2**	Strongly disagree								Strongly agree
	1	2	3	4	5	6	7	8	9
					4/39 (10%)	1/39 (3%)	7/39 (18%)	14/39 (36%)	13/39 (33%)
**Q4**	Evaluate the presence of large amounts of pericardial effusion and tamponade features with POCUS-echocardiography.								
**A4**	Strongly disagree								Strongly agree
	1	2	3	4	5	6	7	8	9
							5/39 (13%)	7/39 (18%)	27/39 (69%)
**Q5_1**	If POCUS-echocardiography is normal and A-line + long sliding is seen in LUS, but the patient’s dyspnea is continued, auscultation is performed to check COPD, asthma, etc., and additional tests are considered if necessary to confirm other metabolic causes.								
**A5**	Strongly disagree								Strongly agree
	1	2	3	4	5	6	7	8	9
		1/39 (3%)			3/39 (3%)	3/39 (8%)	6/39 (15%)	12/39 (31%)	14/39 (36%)
**POCUS-lung ultrasound**									
**Q7**	POCUS-lung ultrasound should be included within this protocol.								
**A7**	Strongly disagree								Strongly agree
	1	2	3	4	5	6	7	8	9
						3/39 (8%)	4/39 (10%)	5/39 (13%)	27/39 (69%)
**Q8_1**	POCUS-lung ultrasound scans six areas of the chest (see attachment 2) but considers adding dorsal if six areas are normal or highly suspected of pneumonia.								
**A8**	Strongly disagree								Strongly agree
	1	2	3	4	5	6	7	8	9
	1/39 (3%)		2/39 (5%)		4/39 (10%)	2/39 (5%)	18/39 (46%)	6/39 (15%)	6/39 (15%)
**Q9**	Check if A-line and Lung sliding are present with POCUS-lung ultrasound.								
**A9**	Strongly disagree								Strongly agree
	1	2	3	4	5	6	7	8	9
						1/39 (3%)	6/39 (15%)	7/39 (18%)	25/39 (64%)
**Q10**	The presence, distribution, and density of B-line are evaluated with POCUS-lung ultrasound.								
**A10**	Strongly disagree								Strongly agree
	1	2	3	4	5	6 (10%)	7	8	9
					1/39 (3%)	4/39 (10%)	5/39 (13%)	9/39 (23%)	20/39 (51%)
**Q11**	Evaluate the presence or absence of pleural effusion with POCUS-lung ultrasound.								
**A11**	Strongly disagree								Strongly agree
	1	2	3	4	5	6	7	8	9
					1/39 (3%)	1/39 (3%)	4/39 (10%)	11/39 (28%)	22/39 (56%)
**Q12**	Evaluate the presence or absence of consolidation with POCUS-lung ultrasound.								
**A12**	Strongly disagree								Strongly agree
	1	2	3	4	5	6	7	8	9
	1/39 (3%)		1/39 (3%)	1/39 (3%)	3/39 (8%)	5/39(13%)	8/39 (21%)	8/39 (21%)	12/39 (31%)
**Q13**	Evaluate the presence or absence of pneumothorax with POCUS-lung ultrasound.								
**A13**	Strongly disagree								Strongly agree
	1	2	3	4	5	6	7	8	9
					1/39 (3%)	2/39 (5%)	8/39 (21%)	10/39 (26%)	18/39 (46%)
**Q14**	The status of lung is evaluated by findings such as B-line, lung sliding, pleural irregularities, pleural effect, and consolidation with POCUS-lung ultrasound.								
**A14**	Strongly disagree								Strongly agree
	1	2	3	4	5	6	7	8	9
				1/39 (3%)	1/39 (3%)	2/39 (5%)	6/39 (15%)	8/39 (21%)	21/39 (54%)
**Other POCUS or additional imaging tests**									
**Q16_1**	If you have RV dilatation or strain with POCUS-echocardiography, but you see a B-line in LUS, there is no lung sliding, consider additional w/u								
**A16**	Strongly disagree								Strongly agree
	1	2	3	4	5	6	7	8	9
				2/39 (5%)	4/39 (10%)	3/39 (8%)	13/39 (33)	9/39 (23%)	8/39 (21%)
**Q17_1**	Aorta CT is considered if aortic detection is suspected if there is a large amount of pericardial effect and tamponade feature with POCUS-echocardiography.								
**A17**	Strongly disagree								Strongly agree
	1	2	3	4	5	6	7	8	9
	1/39 (3%)		2/39 (5%)	1/39 (3%)	2/39 (5%)	2/39 (5%)	5/39(13%)	10/39 (26%)	16/39 (41%)
**Q18**	If you have any additional comments regarding POCUS, please write them down. (Description)								
**A18**	It takes a lot of time to scan six areas to find pneumonia in the field, and there is concern about the efficiency of the extra dorsal scan.								
	Additional w/u for chronic RV strain is not likely to be implemented in the field.								
	On second thought, looking to the dorsal side in POCUS-lung ultrasound is considered to be a waste of time and excessive.								
	POCUS should be carried out in the event of an infection disaster, should be given value as a means to quickly diagnose and treat in the quarantine area.								
	It is difficult to understand the expression that B-line is visible in the US, there is no lung sliding, or wheezing in the stethoscope. Change it to a clear expression.								
	When you present the protocol in the future, you will need a helper to scan your back in a severely ill patient.								
	It would be better to present it only for the US test except for the stethoscope findings.								
	Not adding the dorsal side, can be considered without implementation, so I think this can be logically natural regardless of the actual clinical application.								
	I wonder how we can evaluate this later.								
	If you do a back test, it will take a long time and the accuracy will be low, so it would be better not to do a back ultrasound.								

## Data Availability

This article includes all the data presented in this study.

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
