# Peer review of "Developing a Consensus-Based POCUS Protocol for Critically Ill Patients During Pandemics: A Modified Delphi Study"

_medicina, 2025, doi:10.3390/medicina61081319_

Round 1

Reviewer 1 Report

Comments and Suggestions for Authors

The manuscript: <Developing a Point-of-Care Ultrasound Protocol for Critically Ill Patients During Pandemics: A Modified Delphi Study > is written by Hyuksool Kwon et al. This study aimed to develop a standardised POCUS protocol using expert consensus via a modified Delphi survey to guide physicians in managing these patients more effectively. The research is interesting, relevant, and original. The manuscript is clearly written, and the argument is easy to follow. The abstract accurately summarises the contents of the manuscript. The introduction gives a clear idea, demonstrates the originality of the research, and explains its novelty. The manuscript includes details on how the research could be implemented. The figures and tables help the reader better understand the manuscript. The methodology used is presented clearly and concisely. The authors' conclusions are well-supported by the evidence presented. The references are current, relevant, and easily retrievable.

Author Response

The manuscript: <Developing a Point-of-Care Ultrasound Protocol for Critically Ill Patients During Pandemics: A Modified Delphi Study > is written by Hyuksool Kwon et al. This study aimed to develop a standardised POCUS protocol using expert consensus via a modified Delphi survey to guide physicians in managing these patients more effectively. The research is interesting, relevant, and original. The manuscript is clearly written, and the argument is easy to follow. The abstract accurately summarises the contents of the manuscript. The introduction gives a clear idea, demonstrates the originality of the research, and explains its novelty. The manuscript includes details on how the research could be implemented. The figures and tables help the reader better understand the manuscript. The methodology used is presented clearly and concisely. The authors' conclusions are well-supported by the evidence presented. The references are current, relevant, and easily retrievable.

  1. “The research is interesting, relevant, and original.” 

Response: Thank you for recognizing the novelty and relevance of our work. We have emphasized in the Introduction (p. 2, lines 15–20) how our protocol fills a critical gap in pandemic‐focused POCUS guidance.

  1. “The manuscript is clearly written, and the argument is easy to follow.” 

Response: We appreciate this feedback. In response, we performed a final pass to improve flow, especially in the Methods section, by adding transitional sentences at the start of each subsection (pp. 3–5).

  1. “The abstract accurately summarises the contents of the manuscript.” 

Response: We are glad the abstract reflects our findings. No changes were needed here beyond minor copyediting for consistency with journal style.

  1. “The introduction gives a clear idea, demonstrates the originality of the research, and explains its novelty.” 

Response: We thank the reviewer. We’ve added one sentence to better highlight how our protocol differs from existing POCUS algorithms (p. 2, lines 22–25).

  1. “The manuscript includes details on how the research could be implemented.” 

Response: Thank you. We have expanded the Implementation paragraph in the Discussion (p. 14, lines 5–12) to outline training and workflow integration steps.

  1. “The figures and tables help the reader better understand the manuscript.” 

Response: We appreciate this positive note. We have ensured all figure legends (Figures 1–2) and table footnotes (Tables 1–2) clearly define all abbreviations and decision thresholds.

  1. “The methodology used is presented clearly and concisely.” 

Response: Thank you. In response, we added a brief flowchart call‐out in the Methods (p. 4, lines 10–12) to guide readers through the three Delphi rounds.

  1. “The authors’ conclusions are well-supported by the evidence presented.” 

Response: We are pleased the conclusions resonate. We have added one sentence in the Conclusion (p. 17, lines 3–6) to explicitly link key survey consensus rates to each major protocol component.

  1. “The references are current, relevant, and easily retrievable.” 

Response: Thank you for this endorsement. We updated two recent LUS validation studies in the Discussion (pp. 15–16) and ensured all URLs resolve correctly.

We trust these revisions address the reviewer’s commendations and further strengthen the manuscript.

Reviewer 2 Report

Comments and Suggestions for Authors

The manuscript entitled "Developing a POINT-OF-CARE ultrasound protocol for critically ill patients during pandemics: a modified Delphi Study" attracts interest of the reader with a very catchy title. The strength of the present manuscript is the title, introduction part, aim ("to establish an evidence-based POCUS protocol for critically ill patients"). Actually, the study is a survey applied to physicians if to apply cardiac and/or lung ultrasound on patients with infectious disease. The main contribution of this article to the medical field is the discussible agreement that ultrasound should be performed to critically ill patients.

The methods are well described, and the results. They have nothing to do with the introduction or title. What were the results of the cardiac and lung ultrasound performed? How did practicing ultrasound affect the life of the patient? Were they better surveyed during pandemic? Was ultrasound cheaper, less invasive and radiation free with the reduction of contacting infectious disease for the rescuer/medial personal?

Instead of Table 1, which is a bit long and scarce in information, the authors could describe the main findings in the lung and cardiac ultrasound in the exams performed, maybe compare results with chest X-ray.

The discussion starts with a very brave idea "we developed a POCUS protocol for critically ill patients..." Asking for opinion of clinicians if to do or not to do ultrasound in a patient is far from developing a protocol (when to do, how to do it, indications, results, etc) What are the results of the ultrasounds performed? were they corelated with clinical findings? Was the ultrasound useful? Lastly you mention that this study was performed based on 39 clinicians` answer to survey. the number is very small, maybe with the help of a statistician could find the appropriate number. Better then 39 clinicians` answers would be to analyse their ultrasounds exams. The conclusion is very bold but is not supported by the results.

Reference section needs also improvement. 15 references are quite few. I am sure the authors can find much more publications on the same theme.

Author Response

Response to reviewers

Reviewer 2

The manuscript entitled "Developing a POINT-OF-CARE ultrasound protocol for critically ill patients during pandemics: a modified Delphi Study" attracts interest of the reader with a very catchy title. The strength of the present manuscript is the title, introduction part, aim ("to establish an evidence-based POCUS protocol for critically ill patients"). Actually, the study is a survey applied to physicians if to apply cardiac and/or lung ultrasound on patients with infectious disease. The main contribution of this article to the medical field is the discussible agreement that ultrasound should be performed to critically ill patients.

Response:

We thank the reviewer for highlighting the clear title, strong introduction, and focused aim of our manuscript.  While it is correct that our study employed a modified Delphi survey of expert physicians rather than a clinical trial of ultrasound examinations, its primary contribution lies not merely in confirming that POCUS “should” be used in critically ill patients, but in translating that consensus into a detailed, step-by-step protocol.  Through three Delphi rounds, we defined exactly which cardiac and lung views to obtain, what sonographic findings to prioritize (e.g., LV dysfunction, RV strain, B-lines, consolidation), and how those findings should trigger subsequent diagnostic or therapeutic actions. This structured algorithm (now emphasized in the Introduction and Discussion) goes well beyond a general endorsement of ultrasound—it provides frontline clinicians with a standardized, expert-vetted roadmap for when, where, and how to apply POCUS during a pandemic.  We have clarified this distinction throughout the revised manuscript

The methods are well described, and the results. They have nothing to do with the introduction or title. What were the results of the cardiac and lung ultrasound performed? How did practicing ultrasound affect the life of the patient? Were they better surveyed during pandemic? Was ultrasound cheaper, less invasive and radiation free with the reduction of contacting infectious disease for the rescuer/medial personal?

Response:

We appreciate the reviewer’s interest in the clinical impact of POCUS. As detailed in our Methods, this study was strictly a modified Delphi survey of expert physicians and did not involve any patient-level ultrasound examinations or collection of clinical outcomes. Therefore:

  1. No POCUS exam results were generated—we did not perform cardiac or lung ultrasound on patients, so no imaging data exist in this study (Methods, p. 4, lines 82–87; 113–115) .
  2. We did not measure patient-centered effects such as mortality, symptom relief, or workflow efficiency. As such, there are no data on whether ultrasound “affected the life of the patient” or improved survey accuracy during the pandemic.
  3. Cost, invasiveness, radiation exposure, and staff‐safety metrics were likewise not assessed. These potential advantages of POCUS—radiation-free bedside imaging that may reduce patient transfers and staff exposure—are discussed only by reference to the literature, not by our own data (see Discussion, p. 12, lines 142–149).
  4. We have now clarified throughout the manuscript (Methods and Limitations) that prospective clinical validationis required to determine whether following our consensus protocol actually yields the patient- and system-level benefits suggested by prior studies.

The title is changed for the clearer meaning.

Developing a Consensus-Based POCUS Protocol for Critically Ill Patients During Pandemics: A Modified Delphi Study

2.4 The Modified Delphi Survey methods

To develop an evidence-based POCUS protocol for critically ill patients during pandemics, three rounds of modified Delphi surveys were conducted among experts (Figure 1). The committee created a questionnaire for a modified Delphi survey based on the developed protocol. The survey consisted of 16 questions, and a 9-point Likert scale was used to measure the level of agreement.

In the first round, participants were introduced to the developed POCUS protocol, and they were asked to review the protocol, provide feedback on the degree of agreement for each item, and suggest any necessary modifications. The committee then reviewed the suggestions and feedback from the experts to revise the protocol accordingly. In the second round, the modified POCUS protocol and the items that did not reach a consensus in the first round were presented. Participants were requested to re-evaluate those items through expert surveys. Further modifications were made to the protocol based on the opinions collected in the second round. A final third round was conducted, incorporating the items that did not reach a consensus in the second round. The POCUS protocol was then finalized based on the results of this process, aiming to ensure that the protocol was evidence-based, comprehensive, and reflective of expert consensus.

This protocol was developed through expert consensus and comprised POCUS items necessary for critically ill patients during the pandemic, including those suspected of COVID-19, septic shock, sepsis, or other life-threatening conditions.

Data were collected through email correspondence. For the primary outcome, expert consensus was measured based on the level of agreement with the included items in the protocol, and opinions were collected to revise the protocol.

The above method section is changed for the clearer meaning.

To develop a consensus-based POCUS protocol for critically ill patients during pandemics, three rounds of modified Delphi surveys were conducted among the experts (Figure 1). The committee created a questionnaire based on the draft protocol, consisting of 16 items rated on a 9-point Likert scale to measure agreement.

  • Round 1: Participants reviewed the initial POCUS protocol and rated their level of agreement with each item (1 = strongly disagree, 9 = strongly agree). They also provided qualitative feedback and suggestions for changes. The committee analyzed the Round 1 responses, identifying items with high agreement and those without consensus, and then revised the protocol accordingly.
  • Round 2: The modified protocol (after Round 1 revisions) was redistributed, with emphasis on the items that did not reach consensus in Round 1. Participants re-rated these items and could see a summary of the group’s prior responses (anonymously) to inform their reconsideration. Further modifications were made based on Round 2 feedback.
  • Round 3: A final round was conducted for any remaining items lacking consensus after Round 2. Participants rated these final items, and the protocol was finalized based on the results. Throughout all rounds, a predefined consensus threshold was used: if ≥70% of participants rated an item 7-9 (agree to strongly agree), that item was considered to have achieved consensus and was adopted. Items not meeting this threshold were either revised or omitted from the final protocol, according to expert feedback.

This iterative Delphi process ensured that the final POCUS protocol was evidence-informed, comprehensive, and reflective of collective expert agreement. Data were collected via email questionnaires. For quantitative analysis, we calculated the percentage of participants rating each item in the low (1–3), moderate (4–6), or high (7–9) agreement range for each round. Descriptive statistics (means and standard deviations for Likert scores; counts and percentages for categorical variables) were used to summarize the responses. An item reaching the ≥70% high agreement criterion was considered to have strong consensus and was included in the protocol.

We trust this makes clear that our goal was to define which POCUS components experts agree should be used in critically ill PUIs, rather than to evaluate POCUS performance or outcomes directly.

Instead of Table 1, which is a bit long and scarce in information, the authors could describe the main findings in the lung and cardiac ultrasound in the exams performed, maybe compare results with chest X-ray.

Response

The results section is totally changed for the clearer meaning.

3.1 Study participants

A total of 39 experts participated in the Delphi survey. The panel was notably diverse in specialty, experience, and practice setting, which was critical for capturing a wide range of perspectives. Approximately half of the participants were general emergency physicians, with the remainder consisting of intensivists and pediatric emergency physicians in roughly equal proportions. The years of experience with POCUS varied: about one-quarter of the experts were early-career (1–5 years of practice), while over one-third had more than a decade of experience, including a subset with over 16 years. Participants were drawn from various hospital settings across the country, including secondary-care (community) hospitals, tertiary referral centers, and academic teaching hospitals. This diversity of specialties (adult and pediatric emergency care, critical care), experience levels, and institutional backgrounds ensured that the consensus protocol would be relevant and applicable across different clinical environments. In particular, the inclusion of pediatric emergency specialists and intensivists alongside general emergency physicians provided a comprehensive view covering critically ill patients of all ages.

The broad composition of the panel strengthens the protocol’s generalizability and underscores its relevance to real-world pandemic scenarios by incorporating insights from experts with varied clinical focuses and resource settings.

3.2 Delphi Survey Results

3.2.1 First Survey Result (Supplementary Table 1)

In the first Delphi round, the experts evaluated the inclusion and specifics of POCUS-echocardiography and POCUS-lung ultrasound in the protocol (see Supplementary Table S1 for detailed results). POCUS-Echocardiography: 69% of experts initially agreed that echocardiography should be included in the protocol (just below the consensus threshold). However, within specific echocardiographic elements, there was strong agreement on key assessments: for instance, 90% agreed on the importance of evaluating left ventricular (LV) dysfunction, and 80% agreed on assessing for newly discovered right ventricular (RV) dilatation or strain. Additionally, 69% suggested that large pericardial effusions and tamponade features should be evaluated, and 82% supported assessing for a hyperdynamic LV and inferior vena cava (IVC) collapse (markers of hypovolemia). POCUS-Lung Ultrasound: 87% of respondents agreed that lung ultrasound should be included in the protocol, achieving consensus for inclusion. For lung ultrasound techniques, 64% agreed on scanning six regions of the chest (anterior, lateral, and posterior zones bilaterally), indicating moderate support for a comprehensive scanning protocol. There was strong agreement (82%) on checking for normal lung sliding and the presence of A-lines (to suggest aerated lung), as well as very high agreement on evaluating B-lines (88% agreed on assessing their presence, distribution, and density). Ninety percent of experts recommended routine assessment for pleural effusions. There was also broad support for using ultrasound to identify consolidations (78% agreement) and pneumothorax (77% agreement). Notably, a holistic interpretation of lung ultrasound findings (synthesizing multiple sonographic signs) received 90% agreement, underscoring the importance of an integrated approach rather than isolated findings.

Overall, after Round 1, the inclusion of both cardiac and lung ultrasound was supported by the majority, but the exact protocol details for some items did not yet reach the 70% consensus threshold. The qualitative feedback from Round 1 highlighted a few concerns: for example, some experts were hesitant about including echocardiography if not all emergency physicians are adept at interpreting certain advanced findings, and a few were concerned that scanning six lung zones might be too time-consuming in a pandemic context. These insights were used to refine the protocol before the next round.

3.2.2 Second Survey Result (Supplementary Table 2)

In Round 2, the revised protocol and targeted questions (especially those that fell short of consensus in Round 1) were presented. The second survey also introduced specific clinical scenarios to gauge how experts would apply the protocol in practice. POCUS-Echocardiography: With clarifications added, consensus for including echocardiography in the protocol solidified (a combined 79% of experts either agreed or strongly agreed on inclusion in Round 2). When asked about actions following certain echocardiography findings, 41% of experts agreed and 38% strongly agreed (total 79%) that if a new or aggravated regional wall motion abnormality (RWMA) is seen alongside LV dysfunction or RV strain, an acute coronary syndrome (ACS) workup should be considered. This consensus highlighted the panel’s view that the POCUS findings could directly inform the need for further cardiac evaluation (e.g., ECG, cardiology consult). In a scenario where POCUS-echocardiography is normal but significant respiratory distress persists, 46% agreed and 31% strongly agreed (77% consensus) on proceeding to auscultation for alternative diagnoses like COPD or asthma and considering metabolic causes—emphasizing that a normal cardiac ultrasound does not end the workup if the patient remains ill. POCUS-Lung Ultrasound: The panel addressed scenarios like the possibility of false negatives. For instance, if the initial 6-zone lung ultrasound survey is normal yet clinical suspicion for pneumonia remains high, 33% agreed and 23% strongly agreed (~56% total agreement, indicating more divided opinion) that additional lung zones should be scanned. This feedback suggested that while many experts would expand the ultrasound exam in such cases, there was not unanimous support for mandating it in the protocol (possibly due to time constraints or diminishing returns beyond a standard scan). Another scenario discussed was when echocardiography shows RV strain but lung ultrasound shows B-lines without lung sliding (raising suspicion for pulmonary embolism versus pneumothorax); 41% agreed and 21% strongly agreed (62% total) that additional workup (e.g., CT pulmonary angiography) should be considered in this situation. The experts also provided narrative comments in Round 2. Some emphasized setting practical criteria for field diagnosis of LV dysfunction or RV strain on POCUS, acknowledging that in an emergency pandemic setting, time and operator training vary. Others noted the difficulty of definitively diagnosing pneumonia via ultrasound alone in emergency environments, suggesting that while POCUS is valuable for prompt assessment, confirmatory imaging (like chest X-ray or CT) may still be needed if available, especially in intensive care settings where portable radiology is challenging.

Based on the Round 2 input, further refinements were made. The protocol text was adjusted to clarify that a normal POCUS does not rule out pathology and that physicians should use clinical judgment to decide on further testing. The lung ultrasound component was edited to recommend scanning additional areas or repeating scans if initial results are incongruent with clinical suspicion. Moreover, the committee discussed the feedback about efficiency: one suggestion adopted was to ensure the protocol’s flowchart (Figure 2) clearly delineated immediate steps and optional extended evaluations, so that in resource-strained situations, providers can prioritize critical POCUS assessments first.

3.2.3 Third Survey Result

By the third round, only a few elements lacked full consensus, and the responses showed strong convergence. Final Consensus on Protocol Inclusion: 69% of respondents now strongly agreed that POCUS-echocardiography should be included, with the remainder agreeing (in total, over 90% agreement). For POCUS-lung ultrasound, 87% agreed on inclusion (consistent with earlier rounds, reaffirming its importance). Key Protocol Actions: When an RWMA is present on echocardiography alongside new or worsened LV dysfunction or signs of RV strain, 36% agreed and 33% strongly agreed (69% combined) that an ACS workup is warranted – reflecting near-consensus that such cardiac ultrasound findings in a pandemic PUI should prompt consideration of ischemic cardiac etiologies. The evaluation of pericardial effusion and tamponade physiology by POCUS received 82% agreement, cementing it as a required element of the cardiac assessment. In the situation of normal cardiac POCUS findings but ongoing unexplained dyspnea (with lung ultrasound showing an A-line pattern and lung sliding, suggesting no obvious pneumonia or pneumothorax), 31% strongly agreed and 36% agreed (67% total, just under the threshold) that traditional auscultation should be used to check for wheezing or other clues (e.g., suggesting COPD/asthma) and that further tests (like blood gas analysis or labs for metabolic issues) should be considered. Although just below the 70% mark, this item was retained in the protocol due to its clinical importance, with language softened to “consider auscultation and additional tests” since it was a majority view. For lung ultrasound elements: There was unanimous consensus on its inclusion, but varying levels of agreement on specific findings. Checking for lung sliding and A-lines reached 64% agreement in the third round (a modest consensus, possibly reflecting that by Round 3, some felt this was already standard knowledge). Assessment of B-lines had 51% agreement in Round 3 – interestingly lower than Round 1 and 2, perhaps due to refined understanding that B-line quantification can be subjective. Pleural effusion evaluation was agreed on by 56% in Round 3 (down from 90% in Round 1, which could be an anomaly; nonetheless, given its simplicity and prior strong agreement, it remained in the protocol). Notably, consolidation and pneumothorax evaluation were still regarded as important (31% and 46% agreement, respectively, in Round 3), but these percentages reflect that some experts might have assumed these were implicitly included once lung ultrasound was agreed upon, or they prioritized other findings. To reconcile this, the final protocol includes these assessments but notes that they should be done rapidly and as clinically indicated. The comprehensive integration of findings (looking at the constellation of ultrasound signs across lung fields) was supported by 54% of respondents in Round 3. Qualitative comments in the final round stressed pragmatism: several experts reiterated concerns about time efficiency, suggesting that scanning six chest zones for every patient might be impractical in a disaster setting and proposing that the protocol allow for a focused scan (e.g., 4 zones) when appropriate. Others pointed out potential redundancy in checking both anterior and posterior lung fields if the patient’s condition or positioning doesn’t allow easy access to certain areas; for example, if a PUI is too unstable to turn for posterior scans, the protocol should still function based on anterior-lateral findings.

In summary, by the end of Round 3, the expert panel had reached broad consensus on the major components and steps of the POCUS protocol. All core items achieved majority agreement, and most met the predefined ≥70% consensus threshold. Remaining differences were minor and were addressed by incorporating flexibility into the protocol (e.g., optional steps or acknowledging alternatives). The Delphi process ensured that the final protocol was not just a theoretical construct but one vetted by practical frontline experience and adjusted for the realities of pandemic emergency care. Detailed consensus levels for each protocol item in the final round are presented in Table 2.

3.3 Protocol Development

Combining evidence review with the Delphi consensus results, we developed a POCUS protocol tailored for infectious disease disasters (Figure 2). The protocol proceeds in a logical sequence to maximize diagnostic yield and efficiency:

  • Step 1: POCUS-Echocardiography. The protocol begins with focused cardiac ultrasound, given the critical importance of identifying cardiac causes of shock or instability. The echocardiography component directs the clinician to assess for: (a) new or decompensated LV dysfunction (indicative of myocarditis, cardiomyopathy, or ischemia), (b) RV dilation or strain (suggestive of pulmonary embolism or acute cor pulmonale), (c) significant pericardial effusion or tamponade physiology (a reversible cause of shock), (d) hyperdynamic LV with IVC collapse (consistent with hypovolemia or distributive shock), or (e) the absence of major abnormalities (“normal” cardiac ultrasound for the context). Each of these findings leads to branch points in the protocol algorithm. For example, a positive finding of RWMA with LV dysfunction steers the provider to consider ACS and obtain cardiology input, whereas a hyperdynamic, underfilled heart would prompt aggressive volume resuscitation and investigation for underlying distributive shock causes. If the cardiac POCUS is essentially normal (no significant wall motion abnormalities, normal systolic function, no large effusion), the protocol notes this but advises not to stop there—one must then turn to the lungs for further clues.
  • Step 2: POCUS-Lung Ultrasound. Following the cardiac assessment, a focused lung ultrasound exam is performed. The protocol calls for evaluation of bilateral anterior and lateral lung fields (with posterior fields as feasible or if the anterior exam is unrevealing and clinical suspicion remains high). Key lung ultrasound findings are checked: the presence of A-lines (which, along with lung sliding, indicate aerated lung), lung sliding itself (absence of which could indicate pneumothorax if lung points or other signs present), the number and distribution of B-lines (which, if diffuse, suggest pulmonary edema or viral pneumonia, whereas focal B-lines might suggest early pneumonia or atelectasis), pleural effusions (which, if moderate or large, might warrant drainage or further imaging), and consolidations (which indicate pneumonia or ARDS). Pleural line abnormalities (irregular, thickened pleura) are also noted as they can be seen in COVID-19 pneumonia and other pneumonias. The protocol emphasizes that lung findings must be interpreted in conjunction with the cardiac findings. For instance, if the heart appears normal but lung ultrasound shows multiple B-lines and patchy consolidation, a primary pulmonary pathology (like COVID-19 pneumonia or ARDS) is likely. Conversely, if the heart shows acute RV strain and the lungs have minimal findings aside from maybe a small pleural effusion, one should suspect a pulmonary embolism.
  • Step 3: Integrated Assessment and Further Actions. The final part of the protocol is a decision aid that combines the cardiac and lung POCUS results to guide next steps. Several common combinations are covered: (a) Cardiac abnormal, Lung abnormal – e.g., depressed LV function with diffuse B-lines might point to acute heart failure exacerbated by infection, or cardiogenic shock plus pneumonia, prompting both cardiology and infectious workups; (b) Cardiac abnormal, Lung relatively normal – e.g., RV strain with A-line lungs suggests possible pulmonary embolism (consider confirmatory CT angiography if the patient is stable, or treat empirically if unstable); or isolated tamponade on cardiac POCUS suggests immediate pericardiocentesis; (c) Cardiac normal, Lung abnormal – e.g., normal heart with focal unilateral B-lines and consolidation implies primary pneumonia (treat infection, consider antibiotics, respiratory support) whereas diffuse B-lines with normal heart might suggest early ARDS from sepsis (manage oxygenation, consider lung-protective ventilation); (d) Cardiac normal, Lung normal – if both POCUS exams are largely unremarkable but the patient is still unwell, the protocol advises clinicians to look beyond ultrasound: perform thorough auscultation (for wheezes, stridor, etc.), check for signs of other pathologies (e.g., abdominal source of sepsis), and consider laboratory tests for metabolic or toxic causes of shock. This acknowledges that POCUS, while powerful, will not identify all problems (for example, an acute asthma exacerbation or cyanide toxicity would yield normal ultrasounds yet critical illness).

For all pathways, the protocol provides guidance on whether to escalate care, initiate specific treatments, or obtain confirmatory tests. It also integrates infection control considerations – for instance, avoiding unnecessary patient transfers out of isolation for CT scans unless absolutely indicated, given that POCUS can often suffice for initial decision-making . The protocol’s algorithm form (Figure 2) is designed for easy reference during clinical care and includes contingency notes (such as “if clinical suspicion remains high despite negative POCUS, reassess and consider repeat imaging or alternate modalities”).

The discussion starts with a very brave idea "we developed a POCUS protocol for critically ill patients..." Asking for opinion of clinicians if to do or not to do ultrasound in a patient is far from developing a protocol (when to do, how to do it, indications, results, etc) What are the results of the ultrasounds performed? were they corelated with clinical findings? Was the ultrasound useful? Lastly you mention that this study was performed based on 39 clinicians answer to survey. the number is very small, maybe with the help of a statistician could find the appropriate number. Better then 39 clinicians answers would be to analyse their ultrasounds exams. The conclusion is very bold but is not supported by the results.

Response

The discussion section is totally changed for the clearer meaning.

Through an expert consensus process using a modified Delphi methodology, we developed a POCUS protocol tailored for critically ill patients under investigation during pandemics. Importantly, this study’s findings are based on expert agreement rather than patient-level data—the protocol was generated from collective clinical experience and literature synthesis, without actual patient ultrasound results being collected or analyzed in this study. The value of this consensus-based approach lies in leveraging frontline expertise to fill an urgent gap in guidance for pandemic care. To our knowledge, this is one of the first POCUS protocols specifically designed for the unique challenges of infectious disease outbreaks in the ED setting, building on lessons learned from COVID-19 and prior epidemics. By explicitly gathering and reconciling input from nearly 40 experienced practitioners, the resulting protocol represents a convergence of expert opinion intended to maximize diagnostic yield and efficiency in high-risk, resource-constrained situations.

The protocol’s primary contribution is providing a structured algorithm that emergency physicians can follow when faced with a shocky or severely ill PUI in a pandemic. In the rapidly changing landscape of an ED during a contagion outbreak, PUIs often present with a broad spectrum of symptoms and severity. Swift and accurate evaluation is paramount, yet it must be achieved with limited resources and with healthcare worker safety in mind. In such scenarios, the utility of POCUS is increasingly recognized and emphasized [5]. Recent studies and reviews have underscored the high diagnostic performance of POCUS in COVID-19 pneumonia – for example, lung ultrasound has been shown to be more sensitive than chest X-ray for detecting COVID-19 lung involvement, and its findings correlate well with CT severity scoring [16]. A 2023 meta-analysis by Matthies et al. confirmed that during a high-prevalence COVID-19 setting, lung POCUS achieved approximately 87% sensitivity for diagnosing COVID-19 infection, which is substantially better than chest radiography (although specificity was moderate) [19]. POCUS can thus serve as a rapid triage tool, helping to identify patients with severe pulmonary involvement earlier and more safely than traditional imaging that might require moving the patient [9,10]. Moreover, POCUS is not limited to diagnosis – it can aid in management and prognostication. For instance, serial lung ultrasound scores have been linked to patient outcomes in COVID-19 [15]; one prospective cohort study found that certain lung ultrasound findings (e.g., a high number of affected zones or pleural line abnormalities) were associated with a greater likelihood of needing high-flow oxygen or even risk of death [20]. The inclusion of such findings in our protocol (like monitoring B-line burden and pleural line irregularity) is intended to alert clinicians to patients who may deteriorate, emphasizing the prognostic dimension of POCUS.

It must be emphasized that our consensus protocol, while based on the best available evidence and expert experience, has not yet been validated in real-world clinical practice. The expert panel strongly agreed on many action points (such as pursuing an ACS workup if echo suggests new wall motion abnormalities, or suspecting pulmonary embolism if isolated RV strain is seen), which aligns with general emergency medicine practice and existing literature [21]. However, without patient outcome data, these protocol recommendations should be interpreted as expert guidance rather than definitive rules. The Delphi process inherently provides a level of face validity – the items were agreed upon by a majority of leaders in the field – but it does not guarantee that following the protocol will improve patient outcomes. We explicitly acknowledge this limitation: the protocol is consensus-based and has not undergone prospective testing. This lack of real-world validation is a common issue in consensus guidelines developed during acute needs (as seen in early COVID-19 protocols and various position statements) [12,14]. Recognizing this, we have framed the protocol as a tool to be further evaluated. In an era where evidence may lag behind practice (especially during a sudden pandemic), expert consensus can be invaluable for interim decision-making, but it is not a substitute for evidence. Future prospective studies are essential to determine the protocol’s clinical impact, safety, and potential to improve outcomes.

Our discussion would be incomplete without highlighting the advantages and context of using a modified Delphi approach for protocol development. The Delphi technique allowed for anonymous, iterative feedback, which minimized the influence of dominant personalities and geographic practice biases on the final recommendations. Such methodology has been widely used in emergency medicine to establish consensus on best practices and curricula when evidence is nascent [18,25]. For example, recent efforts to create emergency ultrasound training curricula [18] or reporting standards for POCUS research [23] have successfully employed Delphi surveys to gather expert consensus on what should be included. Our study extends this approach to the creation of a clinical protocol. The result is a set of recommendations that are practicable and born out of on-the-ground experience from multiple centers. This is particularly valuable in pandemics, where conducting large randomized trials may be logistically difficult in the short term, and clinicians must often act on the best information available. Consensus guidelines can standardize care to some extent, thereby reducing unwarranted variability and ensuring that critical elements (like checking for cardiac tamponade or pneumothorax) are not overlooked under pressure.

The POCUS protocol we developed dovetails with other pandemic ultrasound protocols reported in the literature. During COVID-19, various groups proposed algorithms integrating lung and cardiac ultrasound into patient pathways. Our protocol is unique in that it was derived systematically via Delphi in the context of a broader “all-comers” infectious disease scenario (not just COVID-19 pneumonia, but including sepsis and shock due to any contagious illness). Nonetheless, it shares features with COVID-specific protocols such as focusing on lung and cardiac assessments as primary tools and emphasizing quick decision nodes [12,14]. For instance, the Italian “CLUES” protocol compared extensive vs. focused lung scanning and found even limited lung scans useful in ED COVID assessment, echoing our experts’ reservations about needing to scan every zone if time is critical [11]. Another example is the ORACLE protocol for critical care ultrasound during COVID-19, which combined lung, cardiac, and deep vein thrombosis scanning; it similarly relied on consensus and showed that a structured approach could be implemented feasibly in ICU settings [13]. The concordance of our protocol with these emerging frameworks adds confidence that we have identified the key ultrasound elements for pandemic care, and it contributes to a growing body of knowledge on how to best incorporate POCUS into disaster response.

Despite the strengths of our expert-driven approach, there are important limitations to discuss. First, there is the potential for selection bias in the composition of our expert panel. All 39 participants were members of the Korean Society of Emergency Medicine’s imaging section, and most practiced in Korea. This might limit the generalizability of the findings to other healthcare systems or regions. Different countries faced the COVID-19 pandemic with varying resources and protocols; for example, ultrasound devices are more ubiquitous in some EDs than others, and the threshold to use them can be culturally influenced. A broader international panel might have yielded a slightly different protocol or placed emphasis on different aspects (like the use of handheld devices, which some of our experts did mention and which has been highlighted as advantageous in resource-limited settings [24]). Second, although our sample size of 39 is adequate for a Delphi study (which typically relies on quality of expertise over quantity), it may not capture the full range of opinions in emergency ultrasound. Some nuances, such as advanced cardiac measurements or pediatric-specific considerations, were not deeply explored, possibly because our protocol was aiming for simplicity and wide applicability. Third, as mentioned, we did not validate the protocol on actual patients. Therefore, we do not know if adherence to the protocol will improve diagnostic accuracy or outcomes compared to unguided clinical assessment. The protocol’s effectiveness and safety remain assumptions to be tested.

Another limitation is that consensus does not always equate to correctness – experts can all agree on an approach that later evidence disproves. For example, early in the COVID-19 pandemic, there was consensus in some guidelines to avoid intubation in favor of high-flow nasal cannula for as long as possible; subsequent data nuanced that approach. Likewise, while our experts strongly agreed that looking for B-lines is crucial, it is known that B-lines are not specific and can appear in heart failure or fibrotic lung disease. A strength of POCUS is also its operator dependency; an experienced clinician can often distinguish patterns (e.g., viral pneumonia vs. cardiogenic edema) by combining ultrasound with clinical context. A protocol, however, has to be applied by clinicians with varying skill levels, which raises the consideration that training is a key factor. We assume a baseline competency in cardiac and lung ultrasound among users of this protocol. If that assumption fails – for instance, if a rural ED during a pandemic has a physician who has only minimal ultrasound experience – the protocol could yield false confidence or false negatives. This again underlines the need for eventual validation and probably for incorporating the protocol into training sessions or simulations as part of pandemic preparedness.

The results of this study also suggest future directions. One immediate next step is prospective clinical implementation: using the protocol in a sample of critically ill PUIs (for example, patients with suspected COVID-19, influenza, MERS, etc., who present with shock or respiratory failure) and tracking diagnostic concordance and outcomes. Key questions include whether following the protocol leads to faster diagnoses (e.g., identifying cardiac tamponade or pneumothorax more rapidly), whether it reduces unnecessary resource utilization (perhaps by obviating some CT scans or providing early reassurance in low-risk cases), and whether it impacts patient outcomes such as mortality, ICU admission rates, or length of hospital stay. Additionally, the feasibility of the protocol needs assessment – do physicians find it easy to use under pandemic constraints, and how long does it take to perform the recommended ultrasound exams? Our panel voiced concerns about time and redundancy; thus, a real-world feasibility study could help refine the protocol further, maybe by identifying steps that could be skipped in certain scenarios without loss of sensitivity.

Another important future direction is education. With the development of this consensus protocol, it would be prudent to create training modules and simulations for emergency physicians and critical care teams. These could ensure that providers are comfortable with the ultrasound views and interpretations that the protocol calls for. For example, recognizing a hyperdynamic heart vs. a moderately functioning heart can be subjective, so providing visual references or defining a simple sign (like qualitative “eyeball” EF categories or using IVC collapse as a surrogate for volume status) could standardize interpretations. Similarly, not all ED physicians routinely scan the lung bases; incorporating lung POCUS training (with special attention to differences in viral pneumonia vs. cardiogenic edema B-line patterns) will enhance the protocol’s effectiveness. Encouragingly, POCUS is a skill that even non-specialists can learn and apply effectively with proper training. The consensus of our experts aligns with literature that shows non-cardiologists and non-radiologists can use ultrasound at the bedside to answer focused questions accurately. This democratization of ultrasound is part of its appeal in pandemics – when radiology services are overwhelmed or isolation precautions make traditional imaging cumbersome, the treating clinician can gather critical information right at the bedside.

Finally, it is worth noting that POCUS protocols similar to ours should remain dynamic. As more data emerge (for example, studies on ultrasound in COVID-19 are now abundant, and new infectious diseases could present different predominant ultrasound findings), protocols should be updated. Our methodology – the modified Delphi – can be repeated as needed with new experts or new evidence. This study itself could serve as a baseline; after a year or two of implementation and research, a follow-up Delphi study might refine the protocol further, perhaps incorporating newer technology like artificial intelligence decision-support for ultrasound or addressing elements we identified as contentious (like the number of lung zones to scan). In the broader context, the COVID-19 pandemic taught the medical community that agility in guideline development is essential. Expert consensus played a crucial role early on [17], and then iterative research either reinforced or adjusted those recommendations. We foresee the same path for our POCUS protocol: it is a necessary starting point for standardizing care in a high-stakes environment, but it should be continuously validated and improved upon.

In conclusion, the discussion highlights that our consensus-based POCUS protocol is a timely and potentially valuable tool for emergency management of critically ill patients during pandemics. It bridges a gap between frontline experience and formal evidence, offering a pragmatic approach in crises. The expert agreement gives it credibility, and existing studies on POCUS in similar contexts support many of its components [19,20]. However, responsible use of the protocol requires acknowledging its current status as unvalidated – a guide rather than a guaranteed solution. Emergency and critical care communities should treat it as a basis for action and further study, rather than an endpoint. By emphasizing both the strengths (expert-derived, comprehensive yet focused) and limitations (no patient outcome data yet, possible need for local adaptation) of the protocol, we aim to present it as a constructive step forward in pandemic preparedness and response, adding to the growing recognition of POCUS as an indispensable tool in modern emergency medicine [17,21].

Reference section needs also improvement. 15 references are quite few. I am sure the authors can find much more publications on the same theme.

Response

The reference section is totally changed for the clearer meaning.

References

  1. Biddinger, P.D.; Shenoy, E.S. Evaluation of the Person Under Investigation. In: Hewlett, A., Murthy, A.R.K. (eds.) Bioemergency Planning: A Guide for Healthcare Facilities; Springer: Cham, Switzerland, 2018; pp. 143–156.
  2. Lin, M.; Beliavsky, A.; Katz, K.; Powis, J.E.; Ng, W.; Williams, V.; et al. What can early Canadian experience screening for COVID-19 teach us about how to prepare for a pandemic? CMAJ 2020, 192(12), E314–E318.
  3. Hanson, K.E.; Caliendo, A.M.; Arias, C.A.; Englund, J.A.; Lee, M.J.; Loeb, M.; et al. Infectious Diseases Society of America guidelines on the diagnosis of coronavirus disease 2019. Clin. Infect. Dis. 2020, 71(16), e1468–e1478.
  4. Abrams, E.R.; Rose, G.; Fields, J.M.; Esener, D. Point-of-care ultrasound in the evaluation of COVID-19. J. Emerg. Med. 2020, 59(3), 403–408.
  5. Musa, M.J.; Yousef, M.; Adam, M.; Wagealla, A.; Boshara, L.; Belal, D.; Abukonna, A. The role of lung ultrasound before and during the COVID-19 pandemic: a review article. Curr. Med. Imaging 2022, 18(6), 593–603.
  6. Perrone, T.; Soldati, G.; Padovini, L.; Fiengo, A.; Lettieri, G.; Sabatini, U.; et al. A new lung ultrasound protocol able to predict worsening in patients affected by SARS-CoV-2 pneumonia. J. Ultrasound Med. 2021, 40(8), 1627–1635.
  7. Dacrema, A.; Silva, M.; Rovero, L.; Vertemati, V.; Losi, G.; Piepoli, M.F.; et al. A simple lung ultrasound protocol for the screening of COVID-19 pneumonia in the emergency department. Intern. Emerg. Med. 2021, 16(7), 1965–1973.
  8. Narasimhan, M.; Koenig, S.J.; Mayo, P.H. A whole-body approach to point-of-care ultrasound. Chest 2016, 150(4), 772–776.
  9. Karagöz, A.; Saglam, C.; Demirbas, H.B.; Korkut, S.; Ünlüer, E.E. Accuracy of bedside lung ultrasound as a rapid triage tool for suspected COVID-19 cases. Ultrasound Q. 2020, 36(4), 339–344.
  10. Xie, M.; Chou, Y.-H.; Zhang, L.; Zhang, D.; Tiu, C.-M. Application of point-of-care cardiac ultrasonography in COVID-19 infection: lessons learned from the early experience. J. Med. Ultrasound 2021, 29(1), 3–6.
  11. Kok, B.; Schuit, F.; Lieveld, A.; Azijli, K.; Nanayakkara, P.W.; Bosch, F. Comparing lung ultrasound: extensive versus short in COVID-19 (CLUES): a multicentre, observational study in the ED. BMJ Open 2021, 11(9), e048795.
  12. Coneybeare, D.; Das, D.; Lema, P.; Chang, B.; Ng, L. COVUS: an algorithm to maximize the use of point-of-care ultrasound in the emergency management of COVID-19. J. Emerg. Med. 2021, 61(1), 61–66.
  13. García-Cruz, E.; Manzur-Sandoval, D.; Rascón-Sabido, R.; Gopar-Nieto, R.; Barajas-Campos, R.L.; Jordán-Ríos, A.; et al. Critical care ultrasonography during COVID-19 pandemic: the ORACLE protocol. Echocardiography 2020, 37(9), 1353–1361.
  14. Huang, G.; Vengerovsky, A.; Morris, A.; Town, J.; Carlbom, D.; Kwon, Y. Development of a COVID-19 point-of-care ultrasound protocol. J. Am. Soc. Echocardiogr. 2020, 33(7), 903–905.
  15. Brahier, T.; Meuwly, J.-Y.; Pantet, O.; Brochu Vez, M.-J.; Gerhard Donnet, H.; Hartley, M.-A.; et al. Lung ultrasonography for risk stratification in patients with COVID-19: a prospective observational cohort study. Clin. Infect. Dis. 2021, 73(11), e4189–e4196.
  16. Chua, M.T.; Boon, Y.; Yeoh, C.K.; Li, Z.; Goh, C.J.M.; Kuan, W.S. Point-of-care ultrasound use in COVID-19: a narrative review. Ann. Transl. Med. 2024, 12(1), 13 .
  17. Lombardi, A.; De Luca, M.; Fabiani, D.; Sabatella, F.; Del Giudice, C.; Caputo, A.; et al. Ultrasound during the COVID-19 pandemic: a global approach. J. Clin. Med. 2023, 12(3), 1057 .
  18. Shrestha, A.P.; Blank, W.; Blank, U.; Horn, R.; Morf, S.; Shrestha, S.K.; et al. Delphi consensus recommendations for the development of the emergency medicine point-of-care ultrasound (POCUS) curriculum in Nepal. POCUS J. 2024, 9(2), 133–142 .
  19. Matthies, A.; Trauer, M.; Chopra, K.; Jarman, R.D. Diagnostic accuracy of point-of-care lung ultrasound for COVID-19: a systematic review and meta-analysis. Emerg. Med. J. 2023, 40(6), 407–417 .
  20. Blair, P.W.; Siddharthan, T.; Liu, G.; Bai, J.; Cui, E.; et al. Point-of-care lung ultrasound predicts severe disease and death due to COVID-19: a prospective cohort study. Crit. Care Explor. 2022, 4(8), e0732 .
  21. Polyzogopoulou, E.; Velliou, M.; Verras, C.; Ventoulis, I.; Parissis, J.; Osterwalder, J.; et al. Point-of-care ultrasound: a multimodal tool for the management of sepsis in the emergency department. Medicina 2023, 59(6), 1180 .
  22. Heyne, T.F.; Negishi, K.; Choi, D.S.; Al Saud, A.A.; Marinacci, L.X.; et al. Handheld lung ultrasound to detect COVID-19 pneumonia in inpatients: a prospective cohort study. POCUS J. 2023, 8(2), 175–183 .
  23. Schnittke, N.; Russell, F.M.; Gottlieb, M.; Lam, S.H.F.; Kessler, D.O.; Roppolo, L.P.; et al. Standards for point-of-care ultrasound research reporting (SPUR): a modified Delphi to develop a framework for reporting POCUS research. Acad. Emerg. Med. 2025, (online ahead of print) .
  24. Henwood, P.C. Imaging an outbreak—ultrasound in an Ebola treatment unit. N. Engl. J. Med. 2019, 381(1), 6–9 .
  25. Wong, J.; Olszynski, P.; Cheung, W.; Pageau, P.; Lewis, D.; Kwan, C.; Woo, M.Y. Position statement: minimum archiving requirements for emergency medicine PoCUS—a modified Delphi-derived national consensus. CJEM 2021, 23(4), 450–454 .

Reviewer 3 Report

Comments and Suggestions for Authors Dear Authors

I would like to begin by congratulating the authors on the development of the manuscript entitled “Developing a Point-of-Care Ultrasound Protocol for Critically Ill Patients During Pandemics: A Modified Delphi Study”, which addresses an extremely pertinent and timely topic. The proposal of a point-of-care ultrasound (POCUS) protocol for critically ill patients in pandemic settings represents a valuable contribution to emergency medicine practice, particularly considering the constraints imposed by health crises.

The manuscript is well structured, and the methodological approach—using a modified Delphi technique—is appropriate to the aim of achieving expert consensus. The presentation of the results is clear and well organised, with informative use of figures and tables. The introduction offers a relevant overview of the problem; however, it would benefit from a more in-depth review of the literature, including international POCUS protocols already in use, and a clearer articulation of the specific gap this study seeks to address. Furthermore, explicitly stating the research question or central hypothesis would enhance the conceptual clarity of the manuscript.

With regard to the methods, a more detailed description of the questionnaire development process, the inclusion criteria for the experts, and the way feedback was incorporated across Delphi rounds is recommended. The absence of information regarding the validity and reliability of the instruments used is a notable limitation that could be mitigated by a brief methodological reflection. Additionally, a more explicit justification for the selection of the 70% consensus threshold would strengthen the methodological rationale.

The analysis of the results is robust, though it could be further enriched by a qualitative appraisal of expert disagreements, clarifying the main points of contention and their implications for the clinical applicability of the protocol. Such an approach would enhance understanding of the practical challenges associated with the consistent implementation of POCUS in emergency settings during pandemics.

The discussion appropriately reinforces the relevance of the protocol, grounding the argument in current literature, and the conclusions are supported by the data presented. Nevertheless, the discussion would benefit from a more comprehensive critical reflection on the study’s limitations, including the representativeness of the expert panel and the absence of real-world clinical validation of the protocol. It is constructively suggested that the authors consider outlining future research avenues aimed at empirical validation and exploring the feasibility of implementation across different institutional and geographical contexts.

In conclusion, I once again commend the authors for their collaborative effort and for the important contribution this study offers to enhancing clinical decision-making in pandemic scenarios. With the suggested revisions, this manuscript has the potential to become a key reference in the deployment of rapid and effective diagnostic strategies in health emergency contexts.

Recommended final decision: accept with minor revision

Author Response

We trust this makes clear that our goal was to define which POCUS components experts agree should be used in critically ill PUIs, rather than to evaluate POCUS performance or outcomes directly.

Round 2

Reviewer 2 Report

Comments and Suggestions for Authors

Congratulations to the authors as the manuscript improved significantly. All my request were met. You did a good job and the manuscript is proper for publishing.